# Does Green Innovation Improve SME Performance?

**Ni Wayan Rustiarini \*** , **Desak Ayu Sriary Bhegawati and Ni Putu Yuria Mendra**

Accounting Department, Faculty of Economics and Business, Universitas Mahasaraswati Denpasar, Kamboja Street No.11A Denpasar, Denpasar 80231, Indonesia
\* Correspondence: rusti_arini@unmas.ac.id

**Abstract:** The environmental damage phenomenon is a challenge for businesses today, including for small and medium industries in developing countries, such as Indonesia. Green innovation is a solution to answer public concerns over global environmental issues. However, the Small and Medium Enterprises (SMEs) sector generally still focuses on achieving their economic performance. Green innovation is a strategic step for SMEs to increase sustainability and financial performance in the global market. This study aimed to holistically identify the antecedents and consequences when implementing green innovation in SMEs. This study also analyzed the role of green innovation as a mediator in the relationship between intellectual capital, sustainability performance, and financial performance. The survey was conducted on 336 SMEs in Bali, Indonesia. The questionnaire was directly distributed to owners or managers of SMEs over three months. This study proved that intellectual capital positively increased green innovation, SME sustainability, and financial performance. Green innovation was also considered as a mediating variable in the relationship between intellectual capital, sustainability performance, and financial performance. Thus, the implementation of green innovation directs entrepreneurs to fulfill not only social and environmental responsibilities but also encourages SMEs to achieve their economic benefits.

**Keywords:** green innovation; intellectual capital; sustainability performance; SME



## 1. Introduction

Environmental damage is a crucial issue in today's business world. This phenomenon emphasizes the importance of balancing social, economic, and environmental performance. Economic performance is no longer the sole goal of businesses. Entrepreneurs are required to overcome social problems in the community and preserve the natural environment (Bombiak and Marciniuk-Kluska 2018; Butt et al. 2022a, 2022b). Several academicians have also paid great attention to the efforts in promoting sustainability performance through green innovation (Asadi et al. 2020; Li et al. 2022).

Green innovation is the integration of internal intellectual capital of a business with the concept of sustainability. Green innovation is a solution to answer public concerns over global environmental issues (Anik and Sulistyo 2021). The concern on the environment forces entrepreneurs to maximize their internal capacity in creating environmentally friendly innovations, including small- and medium-sized industries. Green innovation reduces the negative environmental impacts caused by their activities (Pacheco et al. 2018) and improves their business financial performance (Novitasari and Tarigan 2022; Przychodzen et al. 2020).

Nevertheless, there are still some research gaps in applying green innovation in the Small and Medium Enterprises (SME) sector, especially in developing countries, such as Indonesia. First, the SME sector generally still focuses on achieving its economic performance (Asadi et al. 2020; Neri et al. 2018), particularly in the short term. This condition causes SMEs to pay less attention to the environmental issues. Adhering to economic goals is not enough to achieve permanent sustainability. SMEs need to improve the performance of social and environmental aspects to achieve long-term economic benefits

(Neri et al. 2018). The development of green innovation is considered a win–win solution to overcome the conflict between economic development and environmental protection (Anik and Sulistyo 2021; Marco-Lajara et al. 2022).

Second, green innovation is a strategic step for SMEs to increase product competitiveness in the global market. Many companies create green innovations to meet strict environmental regulations (Marco-Lajara et al. 2022). Currently, export destination countries, such as those in Europe and America, are increasingly tightening sustainability criteria for products permitted to enter their countries. In addition, SMEs are also asked to provide administrative data related to environmental, social, and corporate governance (Taherdangkoo et al. 2017). Green innovation is the most significant strategy to reduce resource demand and consumption in developing and implementing an effective environmental management system (Asadi et al. 2020). Entrepreneurs are motivated to create environmentally friendly designs and packaging as well as to implement a system focusing on environmental management to reduce waste and pollutions (Marco-Lajara et al. 2022; Song and Yu 2018). Therefore, green innovation is SMEs' proactive reaction to strict government regulations (Taherdangkoo et al. 2017).

The study aimed to holistically identify the antecedents and consequences of implementing green innovation in SMEs. First, this study examined the critical role of intellectual capital in green innovation. Furthermore, the authors analyzed the influence of intellectual capital and green innovation on sustainability and financial performance. In addition, this paper examined the impact of sustainability performance on financial performance. Finally, this study analyzed the role of green innovation as a mediator in the relationship between intellectual capital, sustainability performance, and financial performance.

This study is expected to provide contributions to the development of theories and practices. Theoretically, this study possibly strengthens Institutional Theory, suggesting that every SME can adopt green innovation as a company's proactive strategy in response to global competitions. This study also developed a comprehensive view related to the role of green innovation in mediating the relationship between intellectual capital and SME performance, both sustainability performance and financial performance. From a business practice perspective, this study encourages SMEs to adopt green innovation in SME production activities. In addition, SME owners must improve the employees' knowledge and skills, build green structural-based systems and procedures, and increase networking with external partners implementing green management. Thus, SME managers should maximize the role of intellectual capital to create environmentally friendly innovations.

## 2. Literature Review and Hypothesis Development

### 2.1. Institutional Theory

Institutional theory is one theoretical perspective frequently used in studying green innovation (Li et al. 2022). This theory assumes that institutional pressure requires entrepreneurs to adapt organizational development strategies to the requirements of external institutions. It is undeniable that companies have faced many environmental pressures from various stakeholders (Agudo-Valiente et al. 2017; Garcés-Ayerbe et al. 2019). A business will seek to increase its legitimacy with external isomorphic factors, such as obligations, normalization, and imitation (Qi et al. 2021). Pressure from external institutions encourages SMEs to formulate and implement a company's green innovation strategy (Li et al. 2022).

In SMEs' context, green innovation is a proactive action for SMEs to meet sustainability performance. The dynamic global environment requires SMEs to maximize the potential of human resources to develop green innovation (Anik and Sulistyo 2021). Green innovation includes environmentally friendly product design, pollution prevention, waste recycling, energy-saving technology, and environmental management (Galindo-Martín et al. 2020). The "green" label is an incentive to open new market opportunities, consequently intended to increase performance (Li et al. 2022; Marco-Lajara et al. 2022). Thus, green innovation is a win–win solution to balance the economic, social, and environmental performance (Anik and Sulistyo 2021).

### 2.2. Intellectual Capital and Green Innovation

Environmental damage is a severe challenge for businesses today, including small- and medium-sized industries. Green innovation helps reduce the unfavorable influences possibly caused by the production process (Chen et al. 2018). The concept of green innovation has two main dimensions: green processes and green products (Liu et al. 2021; Ullah et al. 2022). SMEs need to increase their internal capacity, such as intellectual capital, to create green innovation. Human capital enables businesses to adapt to the challenges of sustainable development (Kooli 2020), through the development of green innovation. A dynamic business environment requires employees' knowledge, experience, and skills (Jardon and Dasilva 2017). Firms with sufficient and substantial structural capital will perform many value-creation tasks. Companies with sufficient structural capital and unique processes can improve their innovation performance (Buenechea-Elberdin et al. 2018; Messabia et al. 2022b; Pedro et al. 2018). Companies with sufficient and substantial structural capital can perform many value-creation tasks (Ali et al. 2021a). Relational capital presents interpersonal relationships based on trust and commitment between various interested parties (Sapta et al. 2021a). Companies with high intellectual capital have more innovative competences (Ali et al. 2021a; Arsawan et al. 2022; Marco-Lajara et al. 2022). Therefore, a hypothesis is formulated ass follows.

**H1:** *Intellectual capital positively influences green innovation.*

### 2.3. Intellectual Capital, Sustainability Performance, and Financial Performance

Every business activity should refer to sustainability performance and the integration of economic performance, social environment, and nature. Intellectual capital plays an important role in maintaining a balance between achieving the economic performance, preserving the natural environment, and harmonizing the social environment (Pedro et al. 2018). This intangible asset becomes the primary capital to gain a competitive advantage, considering that this asset has a uniqueness, so it is difficult for competitors to imitate (Aljuboori et al. 2022; Crema and Verbano 2016). Employees' knowledge and skills create innovations (Messabia et al. 2022b) to respond the problems related to environmental pollutions and energy consumptions (Pablo-Romero and Sánchez-Braza 2015; Yusliza et al. 2020). Strong interactions between companies and stakeholders are also considered as effective instruments to collaborate in reducing the negative impacts of their operational activities. Interaction allows organizations to exchange their data sources with external partners in building positive externalities (Ansari et al. 2016). Through innovation and collaboration, intellectual capital aligns sustainability performance with economic performance. Previous research has shown a positive relationship between green intellectual capital and green performance (Marco-Lajara et al. 2022). Likewise, the economic performance supported by employees with competences, skills, and knowledge can produce a competitive advantage, in order to improve an organization's economic performance (Messabia et al. 2022a; Yusliza et al. 2020). Thus, two hypotheses are formulated as follows.

**H2:** *Intellectual capital positively influences sustainability performance.*

**H3:** *Intellectual capital positively influences financial performance.s*

### 2.4. Green Innovation, Sustainability Performance, and Financial Performance

Organizations must take a positive approach to protect the environment from the negative impacts of business operational activities (Abadli and Kooli 2022; Asadi et al. 2020). Green innovation is a strategic step to balance between sustainability and economic performance by creating environmentally friendly products and processes. Green innovation includes two dimensions: green product innovation and green process innovation (Liu et al. 2021; Ullah et al. 2022). Green product innovation aims to select more environmentally friendly raw materials, eliminate harmful substances, and modify product designs to reduce the impact of waste on the environment. Meanwhile, green process innovation aims to reduce energy consumption during the production process and recycle waste into

goods of economic value (Liu et al. 2021; Marco-Lajara et al. 2022; Ullah et al. 2022). Thus, green innovation is the right step to preserve nature to achieve economic profitability. The literature shows that companies use green innovation to reduce production costs and minimize raw material waste (Awan et al. 2021; Mahto and Khanin 2015). Thus, green innovation not only improves the financial and social performance of a business but also reduces the negative environmental impacts caused by its activities (Ullah et al. 2022). Thus, green innovation aligns SMEs' economic interests with the organization's environmental management objectives. Thus, two hypotheses are formulated as follows.

**H4:** *Green innovation positively influences sustainability performance.*

**H5:** *Green innovation positively influences financial performance.*

### 2.5. Sustainability Performance and Financial Performance

The SME sector emphasizes economic components over the other aspects (Van der Byl and Slawinski 2015). However, adhering to the economic goals alone is not enough to achieve permanent sustainability (Neri et al. 2018). The performance of social and environmental aspects is also essential to achieve the associated economic benefits (Asadi et al. 2020; Sapta et al. 2021b). Companies can prevent high social and economic costs due to the environmental damage. Businesses with a sustainability orientation are positively related to business performance, considering that sustainable business practices align with stakeholder preferences that increasingly support sustainability (Kautonen et al. 2020; Landrum and Ohsowski 2018). Entrepreneurs can take advantage of the "sustainability" label as an incentive to open new market opportunities, improving the economic performsance. Previous empirical findings showed that companies engaging in the socially responsible activities experienced healthy financial performance (Cordeiro and Tewari 2015) and improved the company's financial performance (Bahta et al. 2021; Novitasari and Tarigan 2022). A previous literature review revealed that the practice of social responsibility increased market value and company profitability (Soundararajan and Brown 2016). Thus, a hypothesis is formulated as follows.

**H6:** *Sustainability performance positively influences financial performance.*

### 2.6. Green Innovation's Role as a Mediating Variable

The green innovation strategy is created by integrating the intellectual capital with a business's sustainability concept. Intellectual capital provides a better understanding of the background related to the creation of green innovation (Ali et al. 2021b). Entrepreneurs should maximize their intellectual capital to encourage innovation in preventing pollution, save energy, recycle waste, design environmentally friendly products, and implement effective environmental management (Abadli and Kooli 2022; Awan et al. 2021; Marco-Lajara et al. 2022). Companies implementing green innovation have the awareness to fulfill both social and environmental responsibilities as well as increase productivity, efficiency, and cost savings, directly contributing to the competitive advantage and improving the financial performance (Li et al. 2020; Li et al. 2022). Previous empirical studies have shown that green innovation mediated the relationship between green intellectual capital and green performance (Marco-Lajara et al. 2022; Wang and Juo 2021). Therefore, two hypotheses are formulated as follows.

**H7:** *Green innovation mediates the influence between intellectual capital and sustainability performance.*

**H8:** *Green innovation mediates the influence between intellectual capital and financial performance.*

Figure 1 presents the research conceptual framework.

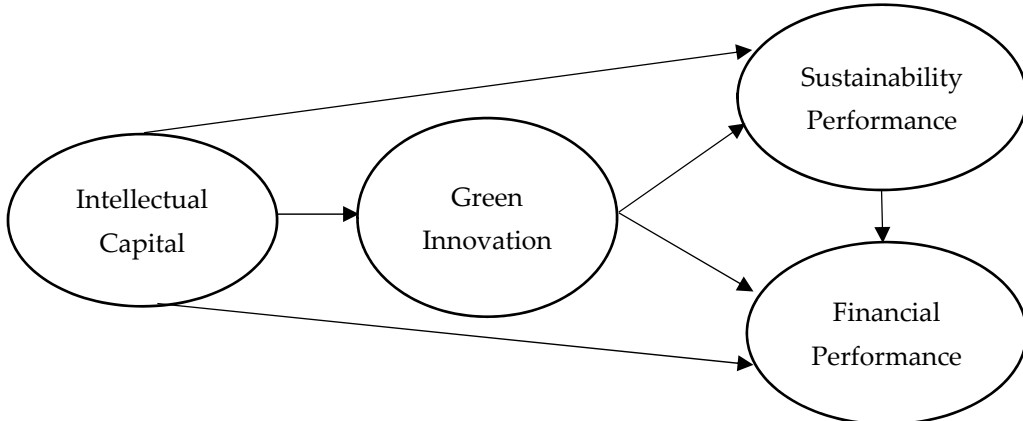

**Figure 1.** Conceptual Framework.

### 3. Methodology

This study involved 336 SMEs in the wood craft industry in Indonesia. The manufacturing sector significantly impacts the environment, yet this industry is reported as being the highest contributor to the environmental problems. This sector uses natural raw materials, while the production process creates waste, possibly damaging nature. Therefore, small- and medium-sized industries need green innovation to promote sustainability performance.

Based on Krejcie and Morgan (1970)'s sampling table, this study selected 336 SMEs meeting the criteria. The first criterion was that the SMEs are considered as a manufacturing industry and still operating actively after the COVID-19 pandemic. Second, the samples belonged to the small and medium enterprises representing the existence of manufacturing SMEs in Indonesia. Thus, this sample is considered to represent the larger population. This study used a questionnaire distributed for three months, from July to September 2022. The questionnaire was directly delivered to the owners or managers of SMEs located in nine districts of Bali, Indonesia.

This study analyzed the variables of intellectual capital, green innovation, sustainability performance, and financial performance. Intellectual capital is an intangible SME resource, including human, structural, and relational capital. Human capital is an employee's characteristics measured through five indicators, including highly skilled (HC1), more creativity (HC2), good experience (HC3), knowledgeable (HC4), and fast in problem-solving (HC5). Relational capital is measured through the characteristics that a business has effective collaboration and intimate communication (RC1), maintains appropriate interaction with stakeholders (RC2), has long-term relationships with customers (RC3), has many excellent suppliers (RC4), and also has a good relationship with a strategic partner (RC5). Structural capital reflects a business condition that has a relevant information system (SC1), efficient operation procedures (SC2), full support for innovation (SC3), easily accessible information system (SC4), flexibility and comfortable culture (SC5), emphasizes new market development (SC6), and has a fast response to changes (SC7). The questionnaire was adapted from the research conducted by Aljuboori et al. (2022) using a five-point Likert Scale.

Green innovation variable consists of green process innovation and green product innovation. Green process innovation is reflected in the growth business harmonizing both economic and environmental benefits (Gproc1), improving the production process promptly in accordance with the consumers' demand for environmental protection (Gproc2) and willingness to invest in the environmental technology development (Gproc3). Green product innovation is reflected in the enterprise's efforts in designing products made from the green and environmentally friendly raw materials (Gprod1), improving production techniques to reduce resource consumption (Gprod2), and creating an efficient and com-

plete waste recycling system. The questionnaire was adopted from previous research (Eiadat et al. 2008; Tu and Wu 2021) using a five-point Likert Scale.

Sustainability performance is the SME performance's balance based on economic, environmental, and social performance. Economic performance includes the SMEs' efforts to involve suppliers in a new product or service development (Econ1), inform the organizational changes influencing the purchasing decision of suppliers (Econ2), and provide information related to the purchasing decisions to all customers. Environmental performance shows the SMEs' efforts to adopt processes, including to reduce water consumption (Env1) and energy consumption (Env2), reduce and recycle waste (Env3), reduce harmful emissions (Env4), and reduce packaging environmental influences (Env5). Social performance reflects the SMEs' efforts to build the employee relationship by providing proper salaries and fair rewards (Soc1), supporting employees to have further education (Soc2), providing a procedure to ensure safe facilities and occupational health (Soc3), and fairly treating employees without any gender or ethnicity discrimination (Soc4). The questionnaire was adopted from the research conducted by Cantele and Zardini (2018) using a five-point Likert Scale.

Finally, financial performance reflects the SMEs' financial performance measured by the number of productions (Perf1), production costs (Perf2), and total net profits (Perf3). The financial performance was measured using a self-assessment method considering that SMEs did not publish their financial reports in public. The questionnaire was adapted from previous research (Cantele and Zardini 2018; Saeidi et al. 2015) and measured using a five-point Likert Scale. This study also examined the role of control variables consisting of business age, employee number, capital, and sales. This study used a variant-based Structural Equation Model based on Smart-PLS to test the research hypotheses.

## 4. Results

### 4.1. Respondents' Demographic Characteristics

The respondents were 336 owners or managers of small and medium enterprises. A proportion of 32.74% of SMEs had been operating for 11–20 years, and their business process had entered the growth age. Thus, SMEs should have paid more attention to green innovation in their production processes. Regarding the number of employees, most SMEs (90.18%) had 1–10 employees. Yet, the number drastically decreased due to the COVID-19 pandemic. Previously, the SMEs employed many employees from local communities. This indicated that SMEs paid more attention to social performance. In addition, most SMEs (86.71%) had between 50 and 500 million rupiahs in their physical assets, excluding land and buildings for business locations. The other characteristic is the average sales of SMEs of 300 million-2.5 billion rupiahs per year (51.79%). This tendency is expected to continuously increase along with the recovery of a country's economic conditions.

### 4.2. Outer and Inner Model Testing Results

Testing using Partial Least Squares (PLS) requires testing both inner and outer models. Testing the outer model involves convergent validity, composite reliability, and Cronbach's alpha. Tests for convergent validity showed a loading value of 0.716 to 0.921, indicating a high correlation with the measured construct. Meanwhile, the value of composite reliability and Cronbach's alpha was more than 0.954. This value indicated that the measured construct met the reliability requirements and had internal consistency. Meanwhile, the Average Variance Extracted (AVE) value was more than 0.663, indicating that these research indicators had adequate convergent validity (Hair et al. 2021).

The measurement of the inner model aimed to measure the structural model evaluated using R-square for the dependent construct. Measuring the structural model to test the influence of intellectual capital, green innovation, and sustainability performance produced an R-square value of 0.384. The measurement model for testing the influence of intellectual capital, green innovation, and financial performance presented an R-square value of 0.387. The R-square value indicated that both models were in the moderate category.

### 4.3. Hypothesis Testing Results

This study had two research models. The first model analyzed the role of intellectual capital and green innovation in sustainability performance. Meanwhile, the second model analyzed the role of intellectual capital and green innovation on financial performance. The results of the direct influence test on these variables are presented in Table 1.

**Table 1.** Results of Direct influence Statistical Test.

| Hypothesis | Construct | Original Sample | T Statistics ($\lvert$O/STDEV$\rvert$) | *p* Value | Information |
|---|---|---|---|---|---|
| H1 | Intellectual Capital -> Green Innovation | 0.235 | 4.737 | 0.000 | Significant |
| H2 | Intellectual Capital -> Sustainability Performance | 0.220 | 4.252 | 0.000 | Significant |
| H3 | Intellectual Capital -> Financial Performance | 0.339 | 6.772 | 0.000 | Significant |
| H4 | Green Innovation -> Sustainability Performance | 0.530 | 13.757 | 0.000 | Significant |
| H5 | Green Innovation -> Financial Performance | 0.152 | 2.609 | 0.009 | Significant |
| H6 | Sustainability Performance -> Financial Performance | 0.289 | 4.941 | 0.000 | Significant |

Table 1 presents statistics in which intellectual capital had a positive influence on green innovation (*p*-value = 0.000), sustainability performance (*p*-value 0.000), and financial performance (*p*-value = 0.000). These statistics indicated that the results of this study supported the formulated hypotheses of H1, H2, and H3. Meanwhile, green innovation also had a positive influence on sustainability performance (*p*-value 0.000) and financial performance (*p*-value = 0.000). This figure indicated that the statistical test results supported the hypotheses of H4 and H5. Testing H6 showed a p-value of 0.009, indicating that sustainability positively influenced financial performance. This study also identified the role of green innovation as a mediator in the relationship between intellectual capital, sustainability performance, and financial performance. Table 2 presents the indirect influence statistical test results.

**Table 2.** Indirect Influence Test Results.

| Hypothesis | Construct | Original Sample | T Statistics $\lvert$O/STDEV$\rvert$) | *p* Value | VAF | Information |
|---|---|---|---|---|---|---|
| H7 | Intellectual Capital -> Green Innovation -> Sustainability Performance | 0.125 | 4.440 | 0.000 | 0.361 | Partial Mediation |
| H8 | Intellectual Capital -> Green Innovation -> Financial Performance | 0.036 | 2.354 | 0.019 | 0.295 | Partial Mediation |

Table 2 presents the two-construct statistical tests. In the first construct test, green innovation was proven to mediate the relationship between intellectual capital and sustainability performance. Likewise, in the second construct test, the statistics showed that green innovation was also proven to mediate the relationship between intellectual capital and financial performance. The statistical test results supported hypotheses H7 and H8. The Variance Accounted For (VAF) values of 36.10% and 29.50%, respectively, indicated that the green innovation variable had a role as partial mediator in both constructs. This study also

examined the role of control variables consisting of business age, employee number, capital, and sales. The statistical tests revealed that those four control variables had no significant impact on sustainability and financial performance. The model's overall test results are presented in Figure 2.

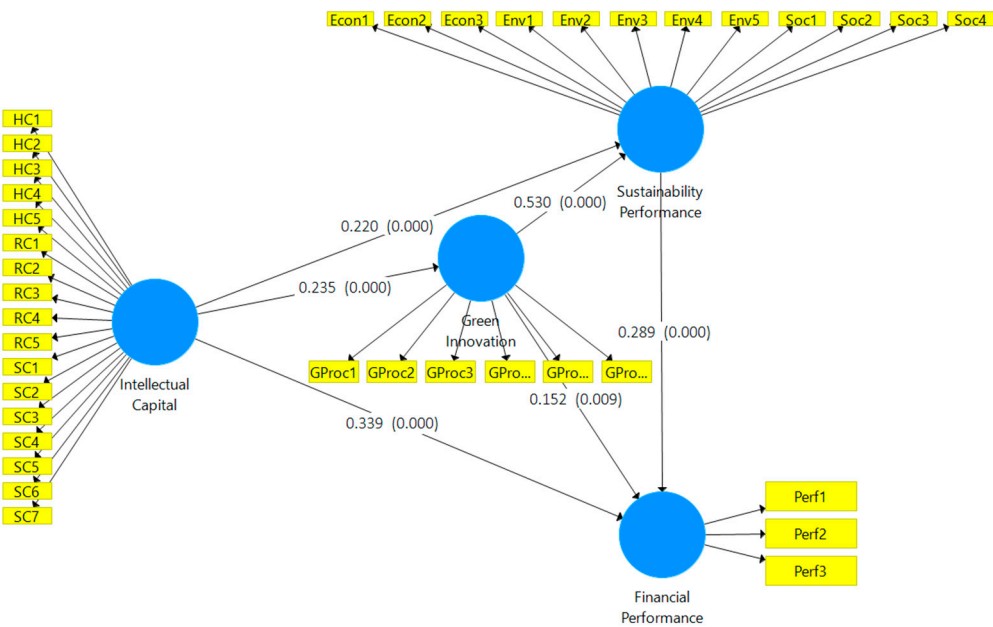

**Figure 2.** Hypothesis Test Results.

## 5. Discussion

The first hypothesis states that intellectual capital positively influences green innovation. The test results supported the hypothesis formulated in this study. The environmental damage phenomenon is, in fact, a large challenge for SMEs today. In this case, SMEs need to maximize the role of intellectual capital as the primary innovation foundation (Ali et al. 2021a). Employees have the knowledge and skills to effectively produce and use green technology in business operations. The employees strive to create processes, operations, and facilities to reduce energy consumption and waste harming the environment (Jirakraisiri et al. 2021). Organizational structures, policies, and culture encourage human resource competencies, leading to increased innovation. In addition, the strong relationship between SMEs and environmentally conscious stakeholders has motivated SMEs to produce green innovations to satisfy the stakeholders. Thus, SMEs having the appropriate intellectual capital will be able to adapt to the challenges of sustainable development (Ali et al. 2021b; Anik and Sulistyo 2021). However, the finding did not support the previous research that intellectual capital had no impact on green innovation performance (Liu et al. 2021).

The statistical test results for the second and third hypotheses revealed that intellectual capital positively influenced sustainability and financial performance. The test results indicated that intellectual capital played an important role in maintaining a balance to achieve the economic, social, and environmental performance. The SMEs, in fact, still focused on achieving economic performance. The concept of sustainability is a challenge for the SME sector to start paying attention to the natural and social environment of the community. In this case, the knowledge and skills of SME employees could create innovations to reduce energy consumption and overcome the environmental pollution problems (Pablo-Romero and Sánchez-Braza 2015; Yusliza et al. 2020). SMEs could collaborate with stakeholders to reduce the negative impacts resulting from their operational activities. Interaction allowed organizations to exchange data sources with external partners to build positive externalities. Thus, intellectual capital directed SMEs to meet sustainability performance while achieving the economic performance (Xu et al. 2021).

The fourth and fifth hypotheses analyzed the influence of green innovation on sustainability performance and financial performance. Based on the statistical test results, green innovation was proven to improve the sustainability and financial performance. Referring to Institutional Theory, entrepreneurs tried to adjust the organizational development strategies with the external institutions' requirements, such as meeting the sustainability criteria for the products they have produced. In the context of SMEs, applying green innovation was a proactive action for SMEs to meet the regulator's pressures (Taherdangkoo et al. 2017). SMEs should develop the environmentally friendly products and processes to substantially reduce the environmental adverse influences. In addition, the creation of products and processes did not only aim to restore the environmental damage, but also improve the SMEs' economic performance (Ullah et al. 2022). SMEs could invest in technology to reduce production costs and minimize raw material waste (Awan et al. 2021; Mahto and Khanin 2015). On the one hand, green innovation could reduce the unfavorable environmental impacts and avoid incidents of environmental destruction (Asadi et al. 2020). These efforts also helped SMEs fulfill their social, environmental, and economic performance. On the other hand, green innovation helped SMEs create positive branding, increasing their position above the average of similar sectors. This act indirectly increased the financial capacity of these SMEs. The results of this research are in line with those of the previous research, stating that green innovation was the right step to preserve nature and achieve economic profitability (Marco-Lajara et al. 2022).

The sixth hypothesis states that sustainability improves financial performance. The statistical test results supported the formulated hypothesis. These results confirmed that social and environmental responsibility became a critical element in achieving the SMEs' financial success in a long term. Businesses with a sustainability orientation were positively related to business performance, considering that sustainable business practices were in accordance with the stakeholders' preferences increasingly supporting sustainability (Kautonen et al. 2020; Landrum and Ohsowski 2018). Entrepreneurs can take advantage of the "sustainability" label as an incentive to open new market opportunities, improving their economic performance. The results of this study, at the same time, strengthened the previous empirical studies, revealing that social responsibility practices increased the market value, company profitability (Soundararajan and Brown 2016), and financial performance, and created a healthy financial performance (Bahta et al. 2021; Cordeiro and Tewari 2015).

Hypotheses seven and eight examine the role of green innovation as a mediating variable. Green innovation is a strategy created by integrating intellectual capital and the concept of business sustainability. Intellectual capital encourages innovation to prevent pollution, save energy, recycle waste, design environmentally friendly products, and implement effective environmental management (Anik and Sulistyo 2021; Marco-Lajara et al. 2022). Intellectual capital also allows SMEs to collaborate with stakeholders to create green innovations. For example, suppliers and consumers with high environmental concerns can collaborate with SMEs to create green products and processes (Xu et al. 2021). SMEs having the awareness to fulfill the social and environmental responsibilities will certainly benefit in relation to cost savings and increased efficiency, as well as directly contributing to financial performance (Li et al. 2022). Thus, the results support the previous empirical studies that green innovation mediated the relationship between green intellectual capital and green performance (Wang and Juo 2021).

## 6. Conclusions

Environmental damage emphasizes balancing the social, economic, and environmental performance in all business sectors. Nevertheless, the SME sector generally still focuses on achieving economic performance. Green innovation is a solution to answer the public concerns on global environmental issues. In addition, green innovation is a strategic step to increase SMEs' sustainability and financial performance. Following institutional theory, entrepreneurs will try to increase their legitimacy when there are external isomorphic

factors, such as obligations. Pressure from external institutions encourages SMEs to adopt green innovations to meet the sustainability performance. This study examined the role of intellectual capital and green innovation on sustainability and financial performance in small and medium industries. The study results showed that intellectual capital positively contributed to increasing green innovation as well as SMEs' sustainability and financial performance. In addition, the green innovation variable also played an important role in mediating the influence of intellectual capital, sustainability performance, and financial performance. Finally, implementing green innovation directed entrepreneurs to fulfill their social and environmental responsibilities and encourage SMEs to achieve their economic benefits.

There are some theoretical and practical implications. When viewed from the theoretical concept, the results of this study supported the Institutional Theory, emphasizing the importance of SMEs adapting their business strategies to the social and environmental responsibilities. This study also highlighted the need for research on antecedent factors influencing the practice of green innovation from the perspective of entrepreneurs. Academicians can collaborate with SMEs to create green innovations. Academicians should increase their number of studies to deeply understand the creation of environmentally friendly processes and products. In addition, academicians can assist SMEs in designing green products and processes.

The practical implications encourage SME owners to optimize their intellectual capital to create environmentally friendly products and processes. The SMEs should accelerate the transition toward a more efficient and responsible energy utilization process. SMEs should consider the use of green innovation and consequences of using this innovation. Furthermore, SMEs should invest some of their profits in creating green innovation and other social and environmental responsibility activities. Thus, this process should provide significant benefits for sustainability and financial performance.

The limitation of this study is that the authors only tested the traditional intellectual capital variable. Theoretically, the traditional intellectual capital perspective is considered not appropriate to examine the environmental issues, due to not having any green dimension. Therefore, further researchers can include green elements in intellectual capital indicators. SMEs with human resources oriented toward green intellectual capital certainly have competencies directly related to environmental aspects, both the natural and social environment.

**Author Contributions:** Conceptualization, N.W.R. and N.P.Y.M.; methodology, N.W.R.; software, N.P.Y.M.; validation, D.A.S.B.; formal analysis, N.P.Y.M.; investigation, D.A.S.B.; resources, N.W.R.; data curation, N.W.R.; writing—original draft preparation, N.W.R.; writing—review and editing, N.W.R.; visualization, N.P.Y.M.; supervision, D.A.S.B.; project administration, N.P.Y.M.; funding acquisition, D.A.S.B. All authors have read and agreed to the published version of the manuscript.

**Funding:** This research and publication were funded by the Ministry of Education, Culture, Research, and Technology through the master contract No. 160/E5/PG.02.00.PT/2022, as well as derivative contracts No. 0967/LL8/Ak.04/2022 and K.888/C.13.02/Unmas/VI/2002.

**Institutional Review Board Statement:** Not applicable.

**Informed Consent Statement:** Not applicable.

**Data Availability Statement:** Not applicable.

**Acknowledgments:** The author would like to thank the research funding provided by the Ministry of Education, Culture, Research, and Technology, and The Research Institute and Community Service of Universitas Mahasaraswati Denpasar.

**Conflicts of Interest:** The authors declare no conflict of interest.

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
