# Peer review of "Does Green Innovation Improve SME Performance?"

_economies, doi:10.3390/economies10120316_

Round 1

Reviewer 1 Report

From the overall presentation I would say that interesting research work has been done. The topic is also important for the readers of the journal. However, I have a few more significant challenges with the paper. 

The research methods used are appropriate but have limitations, and this should be mentioned. The validation of the models could be presented and justified. Furthermore, the uncertainties of the applied analysis could be discussed.   

The methods section is lacking information on the participant recruitment method, namely: a) the recruitment date range (month and year), b) a description of any inclusion/exclusion criteria that were applied to participant recruitment, c) a statement as to whether your sample can be considered representative of a larger population.  

Please include the items (HC1-HC5 RC1-RC5, SC1-SC7 etc) in the case of all variables analyzed (Intellectual Capital, Green Innovation, Sustainability Performance, Financial Performance). 

The results of the structural model analysis should be presented more clearly (Tables or Figures).  Structural research model with path coefficients should be included.  

The novelty of the paper is not sufficiently highlighted by the authors throughout the paper, and in the conclusions section. The novelty of the paper lies only in the geographical context of the research, where research is sparse and limited. You need to improve the practical and academic implications.   

All abbreviates should be explained. 

The quality of Figure 2 is not sufficient. 

Author Response

Dear.

Reviewer 1

Thank you very much for reviewing and providing input on our article.

Here we present the results of the revisions.

Reviewer 2 Report

I would like to thank the authors for this research that aims to holistically identify the antecedents and consequences of implementing green innovation in SMEs. This study also analyzes the role of green innovation as a mediator in the relationship between intellectual capital, sustainability performance, and financial performance.

The research subject is timely, innovative, and highly interesting. It also fits the aim and scope of the journal.

The research is well designed and follows a sound scientific research method.

In order to improve the quality of the research, some adjustments are needed.

Abstract: You need to shortly present the research instrument and number of interviewees.

Several times you used the terms of Business people / business actors; Investors or entrepreneurs are more appropriate than Business people / business actors.

Line 95: 2.2.        Intellectual capital dan green innovation : What do you mean? Anything is missing?

Several sentences need to be verified in order to improve the quality of the research. Example line 97.

Line 131: Check the results of previous research in SMEs. What do you mean by this sentence? It would be better to remove it.

Methodology: You need to mention When the research was realized?

Results: lines 245 and 246:  you need to mention the currency of the mentioned numbers.

The conclusion and implications are insightful; however, you did not mention the limitations of your research. Possibly research on a large scale upon several cities in Indonesia would generate better results.

It would be better to reinforce your literature review with more recent references.

Other minor comments are directly attached to the manuscript.

Author Response

Dear.

Reviewer 2

Thank you very much for reviewing and providing input on our article.

Here we present the results of the revisions.

Round 2

Reviewer 1 Report

In the revised version, the manuscript has been extended and improved. 

Best regards

Reviewer 2 Report

The author made the necessary changes as suggested.